

# Association network analysis identifies enzymatic components of gut microbiota that significantly differ between colorectal cancer patients and healthy controls

Dongmei Ai[1,2], Hongfei Pan[2], Xiaoxin Li[2], Min Wu[2] and Li C. Xia[3]

[1] Basic Experimental Center for Natural Science, University of Science and Technology Beijing, Beijing, China
[2] School of Mathematics and Physics, University of Science and Technology Beijing, Beijing, China
[3] Department of Medicine, Stanford University School of Medicine, Stanford, CA, USA

Corresponding authors
Li C. Xia, l.c.xia@stanford.edu
Dongmei Ai,
aidongmei@ustb.edu.cn

## ABSTRACT

The human gut microbiota plays a major role in maintaining human health and was recently recognized as a promising target for disease prevention and treatment. Many diseases are traceable to microbiota dysbiosis, implicating altered gut microbial ecosystems, or, in many cases, disrupted microbial enzymes carrying out essential physio-biochemical reactions. Thus, the changes of essential microbial enzyme levels may predict human disorders. With the rapid development of high-throughput sequencing technologies, metagenomics analysis has emerged as an important method to explore the microbial communities in the human body, as well as their functionalities. In this study, we analyzed 156 gut metagenomics samples from patients with colorectal cancer (CRC) and adenoma, as well as that from healthy controls. We estimated the abundance of microbial enzymes using the HMP Unified Metabolic Analysis Network method and identified the differentially abundant enzymes between CRCs and controls. We constructed enzymatic association networks using the extended local similarity analysis algorithm. We identified CRC-associated enzymic changes by analyzing the topological features of the enzymatic association networks, including the clustering coefficient, the betweenness centrality, and the closeness centrality of network nodes. The network topology of enzymatic association network exhibited a difference between the healthy and the CRC environments. The ABC (ATP binding cassette) transporter and small subunit ribosomal protein S19 enzymes, had the highest clustering coefficient in the healthy enzymatic networks. In contrast, the Adenosylhomocysteinase enzyme had the highest clustering coefficient in the CRC enzymatic networks. These enzymic and metabolic differences may serve as risk predictors for CRCs and are worthy of further research.

## INTRODUCTION

The human body harbors hundreds of trillions of microorganisms (*Human Microbiome Project Consortium, 2012*; *Turnbaugh et al., 2007*). These microbial communities,

comprising of bacteria, fungi, archaea, and viruses, are highly complex. These microbes colonize internal and external surfaces, such as mouth, esophagus, stomach, colon, respiratory tract, genitourinary tract, and skin (*Grice et al., 2009*). In particular, the disruption of gut microbiota has been linked to a number of gastrointestinal (GI) diseases (*Cotter, Ross & Hill, 2013*). With the rapid development of high-throughput sequencing technologies, the metagenomics approach allows one to extensively investigate the human microbiota in its naturally occurring state. These findings will ultimately lead to a better understanding of the gut microbiota in relation to the onset of GI diseases (*Sung et al., 2016*).

Gut microbiota was known to play an active role in gut homeostasis for many years. The gut microbial composition and metabolic activity can modify the host's susceptibility to diseases via diverse pathologies. For instance, the disruption of gut microbial communities was implicated with CRC (*Cipe et al., 2015*) as the colon microbiota promotes CRC by eliciting pathogenic host responses (*Cho & Blaser, 2012*). In particular, the alterations in the composition, distribution, and metabolism of colon microbiota were shown to disrupt normal homeostasis and lead to the onset of inflammation, dysplasia, and eventually cancer (*Abreu & Peek, 2014*; *Nistal et al., 2015*). More recently, many specific species whose abundance were positively associated with CRC have been identified. Those included *Streptococcus bovis, Helicobacter pylori, Bacteroides fragilis, Enterococcus faecalis, Clostridium septicum, Fusobacterium* spp., and *Escherichia coli* (*Boulangé et al., 2016*; *Gagnière et al., 2016*).

However, these earlier works have primarily analyzed the species abundance and diversity differences among communities to reveal the associations between gut microbiota and CRC (*Halfvarson et al., 2017*; *Sobhani et al., 2011*). It was only recently, researchers began to study microbes at the functional level. The development of metabolomics has markedly facilitated such endeavors. This is made possible because the microbial abundance change will disrupt microbial enzymes that carry out essential physio-biochemical reactions. Such enzymes, as profiled by metabolomics, thus may predict for human disorders. For examples, in the metabolic disorders, such as IBD or obesity, the changes in gut microbiota cause a state of imbalance in metabolic activity and lead to the diseases (*Chang et al., 2015*; *Tsuda et al., 2010*; *Weir et al., 2013*).

Identifying such systematic enzymatic changes requires a high resolution analysis of microbial metabolic networks. Two primary methods are available to construct enzymatic networks: the constraint-based method (*Borenstein, 2012*) and the topological theory-based method. The constraint-based method requires establishing a series of ecological constraint conditions, which requires iterative approximation of unknown parameters in turn. In contrast, the topological theory-based method constructs the network directly by pulling the annotated metabolic and biochemical reaction pathways from databases and removing redundant information from these pathway data as necessary.

Alternatively, in this paper, we demonstrated an approach to construct association networks by computing the local association coefficients of enzymes involved in biochemical reactions and constructing the enzymatic networks using these statistically significant local associations as edges and nodes. Using such derived enzymatic networks, we studied the differentially presented microbial enzymes and metabolic activities using topological analysis. Applying our approach to a large CRC dataset, we identified that
the relative abundance of the enzymatic components of *Bifidobacterium*, *Clostridium*, *Fusobacterium nucleatum*, *Porphyromonas gingivalis*, and *Eubacterium*, were significantly higher in the adenoma and CRC patients, as compared to the healthy controls.

## MATERIALS AND METHODS

### Cohort, fecal samples and metagenomic sequencing

The metagenomic dataset (*Zeller et al., 2014*) used in this study consists of 156 fecal samples of randomly selected volunteers recruited from the Henri Mondor Hospital (Creteil, France), including 61 healthy people, 42 patients with colorectal adenoma, and 53 CRC patients. The dataset were downloaded from the EBI database, where the detailed data description can be found: https://www.ebi.ac.uk/ena/data/view/PRJEB6070.

### Estimation of enzyme abundance

Raw metagenomics data were translated to functional enzymes using the Kyoto Encyclopedia of Genes and Genomes Ortholog database and the HMP Unified Metabolic Analysis Network 2 (HUMAnN2) software, which is a pipeline for efficient and accurate profiling of abundance in microbial pathways from a community based on metagenomic or metatranscriptomic sequencing data (*Abubucker et al., 2012*). We computed the difference of relative abundances for 2,400 microbial enzymes involved in biochemical reactions between the CRC patients and the healthy controls. The results were standardized to eliminate potential bias in sequencing batches, and to remove any batch effects in enzyme abundances either specific to CRC patients or healthy controls.

HMP Unified Metabolic Analysis Network 2 uses an in-house script to normalize the abundance of enzymes. The script provides a method called "total sum scaling" (TSS) normalization. The relative abundance is then calculated as follows:

$$\mathrm{Ra}_i = \frac{a_i}{\sum\limits_j a_j}$$

where $\mathrm{Ra}_i$ is the relative abundance of the $i$-th enzyme and $a_i$ is its absolute abundance.

### Enzymatic association network construction

We performed enzymatic association network construction using the extended local similarity analysis (ELSA) algorithm. ELSA was designed and developed by *Xia et al. (2011)*, which uses a dynamic programming algorithm to effectively discover potential local and time-delayed associations of time-series and cross-sectional data. The dynamic programming algorithm can find all potential global, local and time-delayed associations between two series and identify the association with the highest similarity score as the maximum association between them. We used a false discovery rate $Q$-value cutoff of 0.05 to assess the statistical significance. The resulting enzymatic association networks representing statistically significant associations within the three environments were visualized by the Cytoscape (*Shannon et al., 2003*) software, in which nodes with blue borders in the three association networks represent enzymes enriched in the healthy controls, while yellow nodes represent enzymes enriched in CRC patients.
The ELSA algorithm has been extensively used in the association network analysis of microbial ecological data with a good performance (*Weiss et al., 2016*). Studies using ELSA have discovered symbiotic relationships among microbes and relationships between microbes and the environment that could not be identified by conventional correlation methods. For instance, *Shade, Chiu & McMahon (2010)* found dynamic mixed associations between bacteria in lakes by using ELSA. *Ki, Ryu & Cho (2018)* examined the relationship between bacterial community structure and odor emissions during the degradation of soil and pig carcasses by using ELSA. This algorithm not only finds delayed associations, but also effectively identifies pairwise global associations between for multivariate series data.

### Network topological analysis

Several topological measures were used in our network analysis, including the clustering coefficient and the betweenness and closeness centralities. The clustering coefficient is a measure of the degree to which nodes in a graph tend to cluster together. The clustering coefficient of the nodes represents their proximity in the network. An increased clustering coefficient correlates with the higher tightness of the cluster involving the node and its neighbors, hence, the node's importance. This feature of clustering coefficient allows its use for the identification of key enzymes. The clustering coefficient is defined as

$$C_v = \frac{2n}{k(k-1)}$$

where $n$ is the number of edges between all the $k$ neighbors of node $v$.

The betweenness centrality is a topological measure that refers to the number of times a node acts as a bridge along the shortest path between two other nodes. As the betweenness centrality of a node increases, the frequency with which the node acts as a "mediator" between other nodes also increases, indicating the importance of the node. In an enzymatic association network, a high betweenness centrality indicates that a node plays an important linking role and is likely an important disease-related enzyme.

We also used the closeness centrality measure, which is the inverse of the distance from the node to all other nodes in the network. More important nodes in a network generally have a higher closeness centrality, because they tend to locate close to the center of the network from the geometrical perspective. We identified the shortest paths within the network using the Kruskal's algorithm as implemented in the R package *plyr*.

### Statistical analysis

We tested for the statistically significant abundance difference of enzymes associated with biochemical reactions of gut microbiota between the healthy controls, and adenoma or CRC patients by the two-tailed Wilcoxon rank-sum test.

## RESULTS AND DISCUSSION

### Analysis of disease-associated enzymes

We identified 157 differentially abundant enzymes among healthy control, adenoma, and CRC patients. We selected 13 enzymes with differentially node degree among
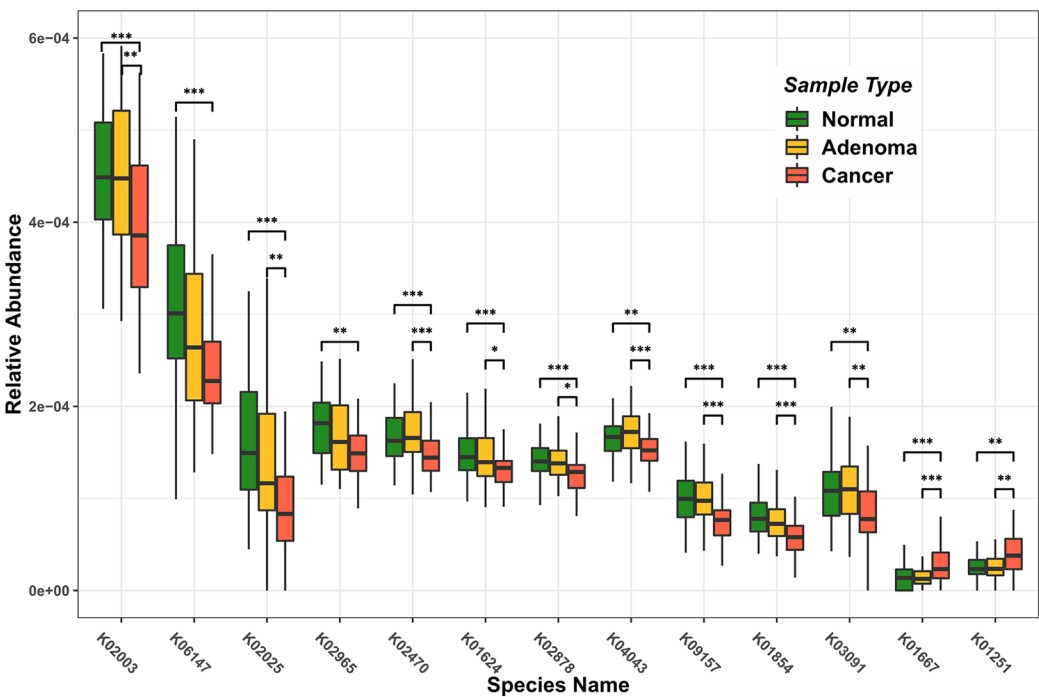

**Figure 1 Enzymes with significant differences of relative abundance in healthy people, adenoma patients, and colorectal cancer patients.** Green bar is the relative abundance of enzymes of healthy people. Orange bar is the relative abundance of enzymes of adenoma patients. Red bar is the relative abundance of enzymes of colorectal cancer patients. ***represent $P < 0.001$, **represent $P < 0.01$, *represent $P < 0.05$.

**Table 1 Enzyme annotations.**

| ID | Name of enzyme |
|---|---|
| K02003 | ABC transport system ATP-binding protein |
| K06147 | ATP-binding cassette, subfamily B, bacterial |
| K02025 | Multiple sugar transport system permease protein |
| K02965 | Small subunit ribosomal protein S19 |
| K02470 | DNA gyrase subunit B |
| K01624 | Fructose-bisphosphate aldolase |
| K02878 | Large subunit ribosomal protein L16 |
| K04043 | Molecular chaperone DnaK |
| K09157 | Uncharacterized protein |
| K01854 | UDP-galactopyranose mutase |
| K03091 | RNA polymerase sporulation-specific sigma factor |
| K01667 | Tryptophanase |
| K01251 | Adenosylhomocysteinase |

healthy control, adenoma, and CRC patients association network. In Fig. 1, we identified 13 enzymes showing statistically significant changes in abundance levels. We conducted a detailed analysis of differential enzymes (IDs and names listed in Table 1) and associated microorganisms. The enzymes K02003 (ABC transport system ATP-binding

protein, $P = 5.22E-04$), K06147 (ATP-binding cassette, subfamily B, bacterial, $P = 9.80E-05$), K02025 (Multiple sugar transport system permease protein, $P = 1.74E-06$), K02965 (Small subunit ribosomal protein S19, $P = 1.17E-04$), K02470 (DNA gyrase subunit B, $P = 3.62E-04$), K01624 (Fructose-bisphosphate aldolase, $P = 2.90E-04$), K02878 (Large subunit ribosomal protein L16, $P = 3.45E-04$), K04043 (Molecular chaperone DnaK, $P = 4.66E-03$), K09157 (Uncharacterized protein, $P = 1.46E-05$), K01854 (UDP-galactopyranose mutase (UGM), $P = 4.31E-06$), K03091 (RNA polymerase sporulation-specific sigma factor, $P = 2.10E-03$) were all significantly higher in healthy controls. The enzymes K01667 (Tryptophanase, $P = 4.57E-04$) and K01251 (Adenosylhomocysteinase, $P = 2.37E-03$) were all significantly higher in CRC patients. These difference trends were also observed between adenoma and CRC patients, however, with less significant $P$-values, suggesting that though adenoma is an intermediary stage in the CRC development, the enzymatic levels observed in adenoma patients resemble more the healthy controls' as compared to that of the CRC patients.

K02003 is the ABC (ATP binding cassette) transporter, and bacteria genomes encoding this protein showed a higher relative abundance in the healthy environment in comparison to either the adenoma or the colorectal environment. As one of the largest metabolic transport systems in humans, the ABC transporter system mainly transfers nutrients, biosynthetic precursors, trace metals, and vitamins, but it also transports lipids, drugs, primary and secondary metabolites. It plays a major role in biosynthetic pathways. It was noted that low ATP-binding cassette protein subfamily (ABCB1, P-glycoprotein) protein levels may promote colorectal carcinogenesis (*Andersen et al., 2013*). ABCB1 protein levels were also found to be lower in CRC tissue as compared to well-differentiated tissue (*De Iudicibus et al., 2008*).

Citing another example, K01854 (UGM, an enzyme that catalyzes chemical reactions) had a higher abundance in healthy samples compared to the other two groups (Fig. 1). This enzyme is specifically involved in galactose metabolism, as well as amino sugar and nucleotide sugar metabolism, and it plays a major role in the production of cellular energy and the modification of proteins and glycolipids. In an earlier study, *Petry & Reichardt (1998)* indicated the importance of galactose metabolism in humans. Galactose can be converted into energy. UGM is one of the main enzymes involved in galactose metabolism. Galactose metabolism is crucial for the health of both neonatal development and adults. Disruption of the production of this enzyme will affect galactose metabolism and even cause transferase-deficiency galactosemia in severe cases. *Brown et al. (2016)* have shown the presence of galactose metabolism disorders in CRC patients. Based on our metabolic association network, we have shown that the abundance of K01854 is lower in the CRC patients than the healthy controls, in agreement with previous findings.

The main enzymatic active genera of galactose metabolism are *Bacteroides* spp. and *Bifidobacterium* spp. The *Bacteroides* spp. participates in polysaccharide degradation and the *Bifidobacterium* spp. participates in galactose metabolism in the gut. Thus, both of these genera may exert protective effects against developing CRC.

There are also enzymes had significantly higher abundances in cancer patients. The most prominent enzymes of the kind were K01251 (adenosylhomocysteinase) and K01667
**Table 2** Topological attributes of 13 nodes of metabolic association networks in the healthy, adenoma, and colorectal cancer environments.

| ID | Clustering coefficient | | | Closeness centrality | | | Betweenness centrality | | |
|---|---|---|---|---|---|---|---|---|---|
| | Healthy | Adenoma of gut | CRC | Healthy | Adenoma of gut | CRC | Healthy | Adenomaof gut | CRC |
| K09157 | 7.42E-01 | 6.70E-01 | 4.72E-01 | 5.66E-01 | 4.62E-01 | 4.14E-01 | 8.76E-03 | 4.58E-03 | 4.62E-03 |
| K06147 | 7.79E-01 | 7.17E-01 | 5.49E-01 | 6.09E-01 | 6.10E-01 | 5.03E-01 | 3.75E-03 | 5.71E-03 | 8.28E-03 |
| K04043 | 7.81E-01 | 7.33E-01 | 5.00E-01 | 5.72E-01 | 5.12E-01 | 4.17E-01 | 3.98E-03 | 9.93E-03 | 1.73E-03 |
| K03091 | 5.85E-01 | 6.65E-01 | 7.20E-01 | 4.97E-01 | 5.31E-01 | 4.95E-01 | 4.32E-03 | 3.07E-03 | 6.63E-03 |
| K02965 | 1.00E+00 | 7.56E-01 | 0.00E+00 | 4.49E-01 | 4.38E-01 | 3.00E-01 | 0.00E+00 | 7.65E-05 | 0.00E+00 |
| K02878 | 9.62E-01 | 6.98E-01 | 0.00E+00 | 4.61E-01 | 5.30E-01 | 3.63E-01 | 9.23E-06 | 2.89E-03 | 2.53E-04 |
| K02470 | 7.31E-01 | 7.18E-01 | 6.43E-01 | 5.97E-01 | 5.35E-01 | 4.70E-01 | 8.81E-03 | 3.79E-03 | 3.20E-03 |
| K02025 | 8.17E-01 | 6.12E-01 | 4.83E-01 | 6.06E-01 | 6.31E-01 | 5.47E-01 | 4.36E-03 | 1.03E-02 | 1.27E-02 |
| K02003 | 8.81E-01 | 7.67E-01 | 8.02E-01 | 5.37E-01 | 5.67E-01 | 4.52E-01 | 1.20E-03 | 3.68E-03 | 7.40E-04 |
| K01854 | 8.76E-01 | 8.07E-01 | 6.19E-01 | 5.31E-01 | 5.59E-01 | 4.45E-01 | 2.72E-03 | 1.65E-03 | 1.84E-03 |
| K01667 | 7.33E-01 | 8.67E-01 | 5.16E-01 | 4.72E-01 | 3.65E-01 | 4.06E-01 | 2.82E-04 | 5.77E-05 | 3.82E-03 |
| K01624 | 9.82E-01 | 7.93E-01 | 7.33E-01 | 5.02E-01 | 5.15E-01 | 3.85E-01 | 2.55E-05 | 1.82E-03 | 7.60E-04 |
| K01251 | 0.00E+00 | 0.00E+00 | 5.90E-01 | 0.00E+00 | 4.38E-01 | 4.03E-01 | 0.00E+00 | 3.69E-04 | 2.32E-03 |

**Note:**
The data range in the table is between (0, 1).

(tryptophanase) (Fig. 1). Therefore, the microbial species producing adenosylhomocysteinase, an enzyme involved in tryptophan metabolism, may facilitate tumor progression as it presented a significantly higher abundance in cancer patients. By using gas chromatography, *Kim et al. (2009)* found adenosylhomocysteinase to be a protein marker for CRC (*Yin et al., 2013*). Interestingly, our enzymatic association network also showed an enriched abundance of K01667 (tryptophanase) in CRC samples. Tryptophanase is a basic amino acid, and tryptophanase metabolism can help cancer escape immune surveillance. Indoleamine 2,3-dioxygenase 1 (IDO1) is a tryptophan-catabolizing enzyme and the main enzyme expressed in malignant inflamed gut. In addition, clinical data have shown that tryptophan metabolism promotes tumor progression (*Santhanam, Alvarado & Ciorba, 2016*).

## Topological attributes of enzymatic association networks

In Fig. S1, we showed the global enzymatic association networks we constructed from the ELSA analysis of the enzymic abundance values. Meanwhile, we exhibited the enzymatic network topological attributes of 13 enzymes in Table 2, including the clustering coefficient, and the closeness and betweenness centralities. As we can see, in the order from healthy, to adenoma and CRC samples, there is a decreasing trend of node degree among the enzymes enriched in healthy controls (nodes with blue circles). Here, the degree of a node refers to the number of neighboring nodes that the node has. Such decreasing trend suggested that these enzymes were highly cooperative in the metabolic processes in health controls (Fig. S1A), while the level of cooperation was reduced in adenoma (Fig. S1B) and further reduced to a minimum level in the CRC environment (Fig. S1C). Notably, a similar decreasing trend was observed in the clustering coefficient (Table 2) of healthy control enriched enzymes, which suggested a gradual loss of importance of these healthy enzymes in a pathogenic CRC gut microbial environment.

The enzyme K02003, an ABC (ATP binding cassette) transporter, had the highest clustering coefficients in the healthy enzymatic network. This ABC transporter was also identified as significantly more abundant in healthy samples by abundance level (see Fig. 1). Both results consistently suggested its status as a signature enzyme species for the healthy gut microbial environment. Similarly, the clustering coefficient of enzyme K02965 (small subunit ribosomal protein S19) was one in the healthy network; however, the coefficient was zero in the CRC network, also suggesting that the enzyme has an important biochemical role in the healthy people, which was no longer found in cancer patients.

In contrast, for the enzymes enriched in the CRC patients, they showed an increasing connectivity from healthy, to adenoma and CRC enzymatic networks. Several such enzymes were uniquely presented in the CRC network. For instance, the K01251 enzyme (Adenosylhomocysteinase), which is a highly connected node in the CRC and adenoma network, however, did not show up in the healthy network at all (Fig. 1), suggesting its functional role in the early stage of tumorigenesis. We also found that the betweenness centrality of K01251 was higher in the CRC environment, and zero in the healthy controls (see Table 2). These results suggest that K01251 may play a pathogenic role in intestinal flora metabolism causing its enrichment in the CRC patients.

## CONCLUSION

We analyzed a large human gut metagenomics dataset by using the HUMAnN2 tool to estimate the relative abundances of microbially produced enzymes in the healthy control, adenoma and CRC patient samples, respectively. We identified differentially abundant enzymes among these healthy, adenoma, and CRC patient groups. We constructed enzymatic association networks representing the healthy, adenoma, and CRC microbial environments using the ELSA algorithm, and analyzed the topological attributes of the resulting networks.

The new enzymatic network analysis approach we took in this study addressed the issue in previous studies that the enzymatic association networks were constructed only for individual metabolic pathways involving a limited number of microorganisms. We were able to globally integrated thousands of enzymes and hundreds of metabolic substrates carrying out biochemical reactions. These microbial enzymatic networks represent the set of essential relationships between enzymes of microbial origin. These relationships were indicators of underlying biochemical reactions and production of metabolites. Among the methods available, only this systematic network-based study can reveal these relationships between microbial communities and their host in high resolution.

The innovative point of our network construction is to use an extensively tested and validated metric—local similarity score for building edges in the network. We know various noises and biases are associated with large-scale network construction, which incur challenges in subsequent topological analysis. Also, many constructed enzymatic networks are not specific simply because of the presence of irrelevant metabolites. The use of an accurate and robust association metrics helped us in truthfully representing these metabolic interactions.

Moreover, the topological attributes of the enzymatic network can carry important information of underlying dynamics between the enzymes and the tumorigenesis process.

For example, the clustering coefficient is a positive indicator of an enzyme's cooperativity with other enzymes. By systematically constructing the networks using the ELSA algorithm and data-mining in node degree, clustering coefficient, betweenness and closeness centralities of the enzymatic network under different cancer stages, we were able to identify signature enzymic species as well as the local shift in network structure of cooperating enzymes as potential cancer risk markers, which merit further research.

## ACKNOWLEDGEMENTS

Dongmei Ai thanks Professor Fengzhu Sun at the University of Southern California.

### Funding

This work was supported by grants from the National Natural Science Foundation of China (61873027, 61370131). Li C. Xia was supported by the Innovation in Cancer Informatics Fund. The funders had no role in study design, data collection and analysis, decision to publish, or preparation of the manuscript.

### Grant Disclosures

The following grant information was disclosed by the authors:
National Natural Science Foundation of China: 61873027, 61370131.
Innovation in Cancer Informatics Fund.

### Competing Interests

The authors declare that they have no competing interests.

### Author Contributions

- Dongmei Ai conceived and designed the experiments, analyzed the data, contributed reagents/materials/analysis tools, prepared figures, and/or tables, authored or reviewed drafts of the paper, approved the final draft.
- Hongfei Pan analyzed the data, prepared figures, and/or tables, authored or reviewed drafts of the paper, approved the final draft.
- Xiaoxin Li performed the experiments, contributed reagents/materials/analysis tools, approved the final draft.
- Min Wu performed the experiments, analyzed the data, contributed reagents/materials/analysis tools, prepared figures and/or tables, authored or reviewed drafts of the paper, approved the final draft.
- Li C Xia conceived and designed the experiments, authored or reviewed drafts of the paper, approved the final draft.

### Data Availability

Data is available at https://www.ebi.ac.uk/ena/data/view/PRJEB6070.

## Supplemental Information

Supplemental information for this article can be found online at http://dx.doi.org/10.7717/peerj.7315#supplemental-information.

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
