# Peer review of "Association network analysis identifies enzymatic components of gut microbiota that significantly differ between colorectal cancer patients and healthy controls"

_PeerJ, doi:10.7717/peerj.7315_

## Round 0.1 · original submission · Major Revisions

Please consider the all comments and concerns the reviewers, more specifically, validation of the results in other datasets which was raised by the two reviewers.

Reviewer 1 ·

Basic reporting

The authors have done a tremendous job in re-writing the manuscript. However, there are still some major concerns that have not been adequately addressed.

First, there are still problems with grammar throughout the manuscript. For example, line 65, “…importantly, however, disruption of gut microbiota has been linked to a number of gastrointestinal.”

Second, on line 62 the authors state, “…gut microbiota…the most capable of influencing human physiology, metabolism, nutrition, and immune function…” This is an assumption. Is there a citation for this statement? As a counterpoint, an infection starts off as having very low abundance but can have an outsourced influence on health and the immune system. Likewise, communities in Cystic Fibrosis are 3-4 orders of magnitude less than the gut but can have devastating immune-mediated effects.

Third, in the abstract on line 54-55, “…changes in microbiota composition may increase the risk of colorectal cancer.” Without formal testing of an odds ratio or similar measure it is hard to accept this claim at face value (i.e. does a higher or lower network centrality lead to lower odds of having an adenoma or carcinoma). This sentence needs to be revised.

Fourth, there are at least 3 papers that have looked at metagenomics and CRC. There has also been a recent meta-analysis in this area. One of these papers (Hannigan et al.) shows that 16S does better than metagenomics in classifying those with and without CRC. The authors need to do a better job of providing a clear rationale for their current study over simply looking for enzymes/compounds.

Fifth, I do not understand how probiotics fit into the introduction or into CRC treatment. Especially since probiotics do not persist in the gut community. Again, the authors are keeping the introduction very broad and use, "disease" rather liberally. This has the consequence of making it hard to understand the main message that they are trying to make. I strongly suggest the authors think carefully over whether they want this presented as a general methods paper or a CRC specific paper.

Sixth, the authors state in a response, “Data standardization is the normalization of data…” I understand that this is what normalization is supposed to do. However, there are many ways to get to this. What approach was used needs to be clearly stated. For example, a Z-score normalization may not improve anything if there is a lot of 0 inflated values.

Seventh, figure 1 is still not easily understandable. This graph is still not clear with respect to what the authors are trying to highlight. I understand what it is they are getting at but this is not clearly shown and raises more questions about the robustness of the analysis if a difference can be found when so many connections are still present.

Eighth, with respect to table 1 this is really what I want a range for. It should be possible if I understand the manuscript to be able to generate either a confidence interval or SD for these measures.

Experimental design

The authors responded to one of my statements with, “The purpose of this project is to detect the differences of enzymes between healthy people and cancer patients, and to deduce the differences of microorganisms from the biochemical reactions, metabolic functions and enzymes involved in biochemical reactions of intestinal microorganisms.” This specific goal that is stated by the authors is not accomplishable with the present data. Having strictly 16S data cannot elucidate the biochemical reactions, metabolic functions, or enzymes involved in the biochemical process. It can infer via known genomes what is likely to be happening. However, this database dependent approach does not necessarily accurately represent the authors’ goals. Additionally, they have not provided enough reasons as to why imputation of metabolite gene abundance is better than 16S given they trace back organisms via the imputed genes to species/genera that would have been identified via 16S.

The authors need to discuss how their method improves upon what was done previously with the data by Zeller et al or reference how it compares to standard practice for processing and analyzing metagenomics

Validity of the findings

Most of the references given by the authors for their algorithm concern dynamic or time-dependent processes. Naturally, the authors need to address how this algorithm performs in identifying novel associations in cross-sectional data (i.e. how does it compare to the original analysis).

In reference to the results presented by figure 3, It is hard to make the case that these enzymes are creating metabolites that are exacerbating CRC. There seems to be a very good association with simply a loss of a specific species (e.g. Bifidobacterium longum). Is it possible for the authors to address or mention that the results could simply be a good marker of the loss of a bacterium rather than the pathway that creates a metabolite of the importance of worsening disease?

Additional comments

Overall, I think the authors have done a tremendous job in improving the manuscript. However, I do not think they have appropriately addressed some of the major concerns that I originally had. I think it would be beneficial for Dongmei and colleagues to think about whether this is supposed to be more of a method or approach-based study or a disease-centered study.

Reviewer 2 ·

Basic reporting

line 65, truncated, should add 'diseases' may be to 'gastrointestinal'
line 106, 'constraint-based method'

Experimental design

the investigation is not rigorous

section 'Topological attributes of metabolic association networks' lines 188-234: it is not clear to me what the authors did here in relation to what is curated from literature, looks more relevant to the discussion section more than results

line 238: 'the standard deviation and mean of 13 enzymes' how were these 13 enzymes extracted ?

starting line 256: it is not clear if the authors did any statistical analysis to compare the abundance of K02003-producing microbes for instance, stacked bar charts could be used for visualization but an appropriate statistical test still needs to be done.

Validity of the findings

lines 184-187 'nodes of which showed enrichment in the colorectal cancer environment, e.g., KI01251
(adenosylhomocysteinase) (Figure 1C), were absent in the metabolic association network of
healthy samples, but present in the association network of the adenoma environment' is there any other kind of high throughput data that can support this finding ? for instance, how about looking for RNAseq data from colorectal patients and healthy subjects, or metabolomics data for colorectal patients ?

Additional comments

lines 107-109 the description of the limitations of constraint-based methods is not very accurate. 'The constraint-based method requires establishing a series of ecological constraint conditions, but this involves unknown parameters and requires performing repeated calculations'. constraint-based methods are based on imposing biologically relevant constraints rather than parameters.

Figure 1: which networks is for healthy subjects vs colon cancer vs adenoma patients ? labels should be indicated in the figure legend as well. Also a more detailed description needs to be added to the legend (e.g. mention details about 1-2 key nodes enriched in healthy vs colorectal patients)

Figure 2: Y-axis scale shows that the abundance is almost scarce, also the relative abundance looks different indeed, are these raw or normalized values ?

I would strongly recommend that the authors try to validate the relative abundance of the enzymes they are focusing on in other datasets. The paper as such presents more of preliminary findings that need to be asserted at other levels.

---

## Round 0.2 · Major Revisions

Please address the issues kindly raised by the reviewer, focusing on comparing the results of your method to those of the authors of the original dataset.

Reviewer 1 ·

Basic reporting

The authors have made some modest improvements to the overall manuscript. I have some concerns that I think are not adequately addressed by the this or previous revisions. Perhaps this iteration of comments will be beneficial in helping the authors consider the pros and cons of the current approach.

Need to correct the title. It is misleading since the authors did not actually perform metabolomics. They imputed gene/enzyme abundances.

There are still minor grammatical errors throughout (e.g. the phrase "may predict for human disorders" in the abstract).

Line 63: microbial communities "causing" colorectal cancer is controversial statement. It has not been shown that the microbiota causes the initial host genetic mutations in key genes like APC or micro-satellite instability. In fact, many of the "microbiota causative" studies require either a chemical or an initial KO of genes for the cancer phenotype to occur. Additionally, the following sentence is one theory as to how the microbiota might contribute to cancer.

Weiss et al. 2015 citation: This paper does not state that eLSA has good performance for the task the authors are using it for. First, this was done on compositional data which Weiss et al. state the algorithm has problems with. The current paper uses a compositional based approach. Second, Weiss et al state that none of the methods are great at detecting interactions between species at that there is room for improvement. It is not clear based on this paper whether eLSA would do better than the previous approach taken by Zeller et al. when using compositional enzyme abundances.

Experimental design

The previous work would have also analyzed the metagenome and presumably also included enzymes that the author's say they focused on. They need to compare their work to what was previously found. Is theirs more believable or was the approach the previous authors took sufficient? It is not clear why this is a better approach than what was already published.

HUMAN2 can also pick out human KEGG annotations. The authors should be reporting how often the KEGG annotations overlapped with human pathways/enzymes.

Validity of the findings

The authors state, “The network topology of metabolic association network exhibited significant difference between the healthy and CRC environments.” It is not clear how the authors can make this call when there is no SD or confidence interval for the three groups. Essentially, the authors are describing an n of 1 experiment if this is based on their network analysis. As I understand the methods, based on the author's responses each sample did not have a network created but they combined them all into a single network.

Using imputed metabolites to show that you have identified bacteria common in metagenomic analysis seems to me to be a little circular. The data set is metagenomics and differences in abundances are known to cause differences in metagenomic analysis. Hence finding key bacteria that the initial data set identified after imputation is almost to be expected and more of a control rather than a reportable novel finding. Using an actual metabolomic data set and finding this would be more convincing that the tools used here would be useful over standard approaches.

Additional comments

Line 57 "undoubtedly" is conjecture.

---

## Round 0.3 · Major Revisions

In addition to addressing the reviewer’s comments, it will be important to test eLSA across additional published datasets.

Reviewer 2 ·

Basic reporting

title: using the word 'enzymatic' twice is redundant. May be just 'network analysis ...etc'

Experimental design

It is still not clear to me how the authors ended up with those particular 13 enzymes. What is the initial number of enzymes tested ? how much were differentially abundant and how much were not ? all these details need to be clearly outlines.

I still have the same concern about the relative abundance values, are these expected to be that low ? what is the distribution of the abundance and relative abundance of all tested enzymes ?

Validity of the findings

the findings presented as such are meant to motivate further investigation but not robust on their own.

---

## Round 0.4 · accepted · Accept

Thank you for adequately addressing the reviewers' comments.

Reviewer 2 ·

Basic reporting

I have no further comments

Experimental design

I have no further comments

Validity of the findings

I have no further comments

Additional comments

the authors have addressed most of the comments I had, thank you